# A Compact, Syringe-Assisted, Vacuum-Driven Micropumping Device

**DOI:** 10.3390/mi10080543

**Published:** 2019-08-17

**Authors:** Anyang Wang, Domin Koh, Philip Schneider, Evan Breloff, Kwang W. Oh

**Affiliations:** 1Department of Electrical Engineering, The State University of New York at Buffalo, Buffalo, NY 14260, USA; 2Department of Biomedical Engineering, The State University of New York at Buffalo, Buffalo, NY 14260, USA

**Keywords:** microfluidics, point-of-care, PDMS (polydimethylsiloxane), micropumping

## Abstract

In this paper, a simple syringe-assisted pumping method is introduced. The proposed fluidic micropumping system can be used instead of a conventional pumping system which tends to be large, bulky, and expensive. The micropump was designed separately from the microfluidic channels and directly bonded to the outlet of the microfluidic device. The pump components were composed of a dead-end channel which was surrounded by a microchamber. A syringe was then connected to the pump structure by a short tube, and the syringe plunger was manually pulled out to generate low pressure inside the microchamber. Once the sample was loaded in the inlet, air inside the channel diffused into the microchamber through the PDMS (polydimethylsiloxane) wall, acting as a dragging force and pulling the sample toward the outlet. A constant flow with a rate that ranged from 0.8 $\mathrm{nl}\cdot s^{-1}$ to 7.5 $\mathrm{nl}\cdot s^{-1}$ was achieved as a function of the geometry of the pump, i.e., the PDMS wall thickness and the diffusion area. As a proof-of-concept, microfluidic mixing was demonstrated without backflow. This method enables pumping for point-of-care testing (POCT) with greater flexibility in hand-held PDMS microfluidic devices.

## 1. Introduction

Highly demanded features in point-of-care testing are autonomy, compactness, and the absence of an external pumping force. In a laboratory setting, manual injection pumps are widely used for microfluidics due to their simplicity. However, manual direct injection is not suitable for devices that require constant and controllable flow rates. Among the various injection methods, the passive pumping method [1,2,3,4,5,6,7,8,9,10,11,12,13,14,15,16] and, especially, the capillary pumping method [5,6,7,8] are widely studied for point-of-care testing for achieving a hand-held lab on a chip-type device. Juncker et al. [5] demonstrated autonomous transportation aliquots of different liquids in sequence using the capillary system. However, the performance of a capillary pumping system depends on surface conditions, fjor example, stable surface conditions are required to achieve constant and controllable flow rates. Recently, a vacuum-assisted pumping method has been introduced. Hosokawa et al. [3,4] demonstrated sample liquid driving methods by using the gas solubility of polydimethylsiloxane (PDMS), where the device was pre-degassed before the operation. However, due to the reabsorption of air once the device was exposed to air, the flow rate decayed immediately. In our previous study [9,10,11,12], a syringe-assisted pumping method which utilized the gas permeability of polydimethylsiloxane was introduced. The pump can drive the liquid without external pumps, however, there are inherent drawbacks. First, the flow rate was significantly reduced after the flow enters the part of channel surrounded by a vacuum chamber. This occurs because the air diffusion area decreases while the fluid approaches the dead-end channel surrounded by the vacuum chamber, resulting in a non-constant flow rate. Achieving a stable flow rate using the passive pumps has been widely studied over the years. Secondly, a pump integrated on the same layer with a fluidic channel, as in the previous reported syringe-assisted pumps [10], is a drawback. The pump height needs to be identical to the fluidic channel unless the users decide to make complicated multiple layers. As the geometry of pump is one of the major design factors for achieving the designated flow rates, the previous reported pump can be limited in controlling flow rates. With the adjustable height design in the proposed pump, the increasing flow rate can be achieved. In addition, it is noteworthy that the proposed pump is designed to be a compact pump, which allows the researchers to operate their designed microfluidic devices just by attaching the pump. In this paper, we propose a compact syringe-assisted vacuum-driven pump with a constant and increasing flow rate for integrated microfluidic networks.

## 2. Materials and Methods 

### 2.1. Theory

The concept of a vacuum-assisted pump involves degassing the vacuum chamber and pulling air out from the microfluidic channel through a gas permeable PDMS thin wall. By creating a pressure difference across the PDMS membrane, the gas diffusion from inside the channel to the vacuum chamber is generated. As a result, the gas pressure inside the channel ($P_{\mathrm{gas}}\left(t\right)$) is always lower than the gas pressure at the other side of the liquid column, which is atmospheric pressure ($P_{\mathrm{ATM}}$). Thus, a loaded sample can be triggered by the generated pressure difference across the liquid column. Previous studies [9,10,11,12] have predesigned vacuum chambers to be on the same layer as the fluidic channel, which caused limitations in making some applications. In this paper, we propose a micropump which can be used for conventionally designed fluidics channel, as shown in Figure 1.

The flow rate *Q* utilizing gas permeability of PDMS is written as [10]: (1)$$Q\left(t\right)\approx k\frac{FS}{C_{\mathrm{ATM}}}=kD\frac{C_{\mathrm{PDMS}}-C_{\mathrm{VAC}}}{C_{\mathrm{ATM}}}\frac{S}{w}\;\to Q\left(t\right)\propto \frac{S}{w}$$
where, k is the empirical factor including the viscous effect of the pumped liquid flow. $D\;(m^{2}\cdot s^{-1}$) is the diffusion coefficient of air in PDMS. $F\;(\mathrm{mol}\cdot m^{-2}\cdot s^{-1})$ is the steady-state air flux diffusing into the vacuum chamber. $S\;(m^{2})$ is the total surface area that allows air to diffuse out of the microchannel, which is defined as the diffusion area. $C_{\mathrm{PDMS}}$, $C_{\mathrm{VAC}}$, and $C_{\mathrm{ATM}}\;\left(\mathrm{mol}\cdot m^{-3}\right)$ are the air concentrations in the PDMS, the vacuum chamber, and the atmosphere, respectively. Here, $C_{\mathrm{ATM}}=44\;\mathrm{mol}\cdot m^{-3}$. w (m) is the PDMS wall thickness. 

An estimation of the characteristic time to allow for the constant air flux was done by Xu et al. [10] and they examined the diffusion time $t_{D}\left(s\right)$ across the PDMS wall:(2)$$t_{D}\approx w^{2}D^{-1}$$

For example, if $D=3.4\times 10^{-9}{\;m}^{2}\cdot s^{-1}$, w=100 μm, the time to initialize the steady-state air flux is $t_{D1}\approx 2.94\;s$. Another characteristic time to diminish the steady-state air flux is determined by the gas diffusion from the vacuum chamber to the surface of the pump. For example, if the proposed pump has a thick PDMS layer (~ 5 mm), the time to diminish the steady-state air flux is $t_{D2}\approx 2\;h$. The air flux will be kept steady within $t_{D1}\ll t\ll t_{D2}$.

The pressure change inside the microfluidic channel can be described using ideal gas laws. An equivalent pressure model is shown in Figure 2.

The molar flux diffusion, $\Delta N_{A\;}$ out of microfluidic channels into a vacuum chamber is written using the permeability coefficient p (Barrer) as:(3)$$\Delta N_{A}=\frac{1}{w}p\left(P_{\mathrm{gas}}\left(t\right)-P_{\mathrm{VAC}}\right)$$
where, $P_{\mathrm{gas}}\left(t\right)\;\left(\mathrm{Pa}\right)$ is the gas pressure inside the channel at a given time and $P_{\mathrm{VAC}}\;\left(\mathrm{Pa}\right)$ is the gas pressure inside the vacuum chamber. The pressure inside the vacuum chamber was calculated using the ideal gas law (see Appendix A). Several papers have reported about the permeability of PDMS [17,18,19,20,21]. Markel et al. [17] measured the permeability of PDMS to various penetrants including N_2_, O_2_, ($p_{N_{2}},\;p_{O_{2}}$, respectively), etc. with a 35 μm thick PDMS layer as a function of pressure at 35 °C. The measured permeabilities are $p_{N_{2}}=400\pm 10$ Barrer and $p_{O_{2}}=800\pm 10$ Barrer despite a change of the transmembrane pressure difference from 101 kPa to 1652 kPa for the pressure on the one side that was kept constant, while on another side the pressure was set to be the atmospheric pressure. Another paper, Lambrti et al. [19], reported the gas permeability as a function of the PDMS mixing ratios (5:1, 10:1, and 20:1, the transmembrane pressure difference at 25, 50, and 75 kPa). The measured gas permeability ranges from 100 Barrer to 3000 Barrer, assuming a constant value of the permeability does not lose accuracy when the pressure difference across the membrane ($P_{\mathrm{gas}}\left(t\right)-P_{\mathrm{VAC}}$) is larger than 50 kPa ($P_{\mathrm{gas}}\left(t\right)-P_{\mathrm{VAC}}>50\;\mathrm{kPa}$). As the pressure difference across the membrane ($P_{\mathrm{gas}}\left(t\right)-P_{\mathrm{VAC}}$) decreases ($P_{\mathrm{gas}}\left(t\right)-P_{\mathrm{VAC}}<50\;\mathrm{kPa}$), the permeability gets smaller. For instance, at $P_{\mathrm{gas}}\left(t\right)-P_{\mathrm{VAC}}=25\;\mathrm{kPa}$, the permeability is nearly half of the value at $P_{\mathrm{gas}}\left(t\right)-P_{\mathrm{VAC}}=75\;\mathrm{kPa}$. Considering that the diffusion coefficient can be described using the permittivity [17], Equation (1) is rewritten as:(4)$$Q\left(t\right)\approx kp\left(P_{\mathrm{gas}}\left(t\right)-P_{\mathrm{VAC}}\right)\frac{1}{C_{\mathrm{ATM}}}\frac{S}{w}$$

Equation (4) describes that the diffusion area and the PDMS wall thickness are the geometrical parameters which change the micropump flow rate.

### 2.2. Device Fabrication

The device was fabricated using soft lithography to form microfluidic channels by PDMS [22,23,24]. First, a mold was fabricated using a 3 inch silicon wafer (University wafers, South Boston, MA, USA). A wafer was submerged into buffered hydrofluoric acid (BHF) at room temperature for 5 min to remove the native silicon dioxide layer. Next, the wafer was cleaned using acetone first, methanol second, and rinsed with deionized water following nitrogen gas blowing. Then, the wafer was placed on a hot plate at 120 °C for 5 min for complete dehydration. In order to reach the 50 μm channel height, negative photoresist SU-8 2050 (SU-8 2050, Micro-Chem Corp, Newton, MA, USA) was spin coated on top of the wafer by using a spin coater (Spin Coater - Brewer Science CEE-200, Rolla, MO, USA. Next, soft bake was conducted at 65 °C for 3 minutes following 95 °C for 9 min. Then, the mask was aligned using the infrared (IR) mask aligner and ultraviolet (UV) was exposed following hard bake at 65 °C for 2 min and 95 °C for 7 min. Then, the wafer was rinsed in SU-8 developer for 5 min to remove the unexposed photoresist following isopropyl alcohol (IPA) rinsing. Finally, the wafer was blown dry by compressed nitrogen air to remove IPA. In order to protect the mold and facilitate releasing PDMS from the mold, release agent hexamethyldisilazane (HMDS), the hydrophobic substancewas coated on the surface by exposing the mold surface to vapor HMDS. After making the mold structure, the PDMS base and curing agent were mixed at the ratio of 10:1, then, degassed in the vacuum chamber to remove air bubbles. Next, the mixture was poured on the mold and baked at 80 °C for 3 h to cure the PDMS. After cooling down the PDMS, the PDMS replica was peeled off the mold and all inlet and syringe ports were punched using biopsy. Then, an oxygen plasma treatment was performed for bonding the PDMS and glass substrate. In order to stabilize the PDMS surface condition, the bonded device was placed on the hot plate and baked at 100 °C for 48 h before testing. 

## 3. Results and Discussion

In order to study the flow rate of the proposed pump, two sets of experiments were performed. 

A traditional microfluidic channel is designed as a bottom layer, while the proposed pump is designed as a top layer to drive the liquid in the bottom layer. In the first set, three pumps with different thicknesses of PDMS wall (100 μm, 150 μm, and 200 μm) were designed keeping the diffusion area at 0.31 mm^2^. In the second set, pumps with different diffusion area (0.31 mm^2^, 0.51 mm^2^, 0.70 mm^2^, and 1.08 mm^2^) were designed keeping the thickness of the PDMS wall as 100 μm. In addition, the effects on flow rate from different diffusion areas were studied by changing the height of pumps. To be specific, by changing the PDMS wall height from 50 μm, 85 μm, and 120 μm, the diffusion area varied from 0.31 mm^2^ to 2.59 mm^2^.

In the first set, (a) schematic of designed pump, (b) tested microfluidics channel, and (c) flow rate testing results are shown Figure 3. In the top layer, the proposed pump consists of a hollow hole with an inner radius of 1 mm surrounded by a PDMS thin membrane with a wall thickness of 100 μm, 150 μm, and 200 μm. In order to prevent the PDMS wall from collapsing, small cylindrical pillars were designed around the hollow hole. The hollow hole was bonded to the outlet (radius is 0.75 mm) of the microfluidic channel in the bottom layer. A straight linear microfluidic channel with 100 μm width, 14 mm length and 50 μm height was chosen as a test channel. The liquid position was measured over time. All of the devices were tested at least three times. As shown by the graph in Figure 3, the pumped volume is proportional to the time, thus, a constant flow rate was achieved despite the liquid position. As described in Equation (1), the flow rate is inversely proportional to the wall thickness and ranges from 0.8 $\mathrm{nl}\cdot s^{-1}$ to 2.6 $\mathrm{nl}\cdot s^{-1}$.

Next, the diffusion area effect on flow rate was studied. The diffusion area of the pump was controlled by changing the shape of the PDMS wall while the PDMS wall thickness was set to be 100 μm (see Figure 4a). First, the PDMS wall height was fixed at 50 μm so that the four different pumps have a diffusion area of 0.31 mm^2^, 0.51 mm^2^, 0.70 mm^2^ and 1.08 mm^2^. The liquid position was measured over time. All of the devices were tested at least three times. As shown by the graph in Figure 4b, the pumped volume was proportional to the time, thus, the pump provided a constant flow rate despite liquid position. The flow rates are proportional to the diffusion areas (see Figure 4c). Then, in addition, the PDMS wall heights were changed (85 μm and 120 μm) in order to further study the diffusion area effect on flow rate (Figure 5). Unlike the device with a pump embedded on the same layer with the microfluidic channel, such as in previous studies [9,10,11,12], the proposed pump is designed independent of a microfluidic channel design. In other words, the PDMS wall height does not need to be the same as the height of the channel, which was a limitation in a previous study [10]. 

As shown in Figure 5, the flow rate is proportional to the diffusion area as described in Equation (1). There are a few possible reasons for the variation of flow rate. As previously studied, the flow rate is independent of the surface condition, however, the surface roughness at the inlet microchannel boundary condition is still critical to determine the burst pressure [25,26], which will affect sample initial velocity. However, the surface roughness or abrupt geometrical change, such as the case caused by dust, is hard to predict. As shown in Figure 3 and Figure 4, the initial flow velocity (t=0 s) is relatively unstable as compared with the flow rate when liquid flows at a certain time (t>10 s). The constant flow rates that are measured are discussed using Equation (4). As the diffusion area and the PDMS wall thickness are the geometrical parameters, the only variables affecting the flow rates are the permittivity and the pressure difference across the membrane:(5)$$Q\left(t\right)\propto p\left(P_{\mathrm{gas}}\left(t\right)-P_{\mathrm{VAC}}\right)$$

As from the two sets of experiments in Figure 3b and Figure 4b, the flow rates were constant while the samples flow inside the channel. Therefore, we believe $p\left(P_{\mathrm{gas}}\left(t\right)-P_{\mathrm{VAC}}\right)$ can assume being constant while the samples flow inside the channel. One of the possible reasons is that the fluidic channel volume (70 nl) is smaller than the outlet volume (e.g., 5 μL), which makes the pressure $P_{\mathrm{gas}}\left(t\right)$ change relatively small while the samples flow inside the channel. Further analysis and simulation are planned for future study. Currently, there are limited reports of efficient methods that have conducted direct measurements on the gas pressure inside the microchannel in the laboratory level, which, therefore, needs to be studied in future research. 

In order to create a proof-of-concept, micromixing is performed using the proposed pump. The micromixing channel design by Zhai et al. [12] is used. One of the advantages of using the proposed device is that the pump can be used for the operation of the device designed by [12]. In the test channel design [12], the device design consists of two inlets, an inlet pressure balance channel for offsetting pressure difference at the inlets, the micromixing channel and the dead-end channel connected to the micropump. The vent and the inlet pressure balancing channel served as an offset to inlet pressure difference between A and B to avoid any backflow while sample mixing [12]. As shown in Figure 6, red dye water and blue water were mixed in the mixing channel. Please refer to the video clip for one of the micromixing demonstrations. It is important to mention that the proposed pump can selectively drive the liquid into the micromixing channel, while the integrated pump can affect the sample flow in the pressure balance channel. The proposed pump has the potential to be used for complicated microfluidic networks.

The operation regime in the proposed pump provided a constant flow rate. The measured flow rates were proportional to the diffusion area and inversely proportional to the PDMS wall thickness. In a previously reported paper, [10], the flow rate was divided into the following two phases: a constant flow rate where the diffusion area is constant, $S_{0},$ and the time dependent flow rate where the diffusion area decreases as the sample reaches the dead-end channel. With regards to the proposed pump, the device operates in a constant flow regime only while the sample flows inside the microchannel since the diffusion area stays constant. The flow rate $Q_{I}\left(t\right)$ and the pumped volume $V_{I}\left(t\right)$ can be rewritten as:(6)$$Q_{I}\left(t\right)\approx k\frac{FS_{0}}{C_{\mathrm{ATM}}}=kD\frac{C_{\mathrm{PDMS}}-C_{\mathrm{VAC}}}{C_{\mathrm{ATM}}}\frac{S_{0}}{w}\;V_{I}\left(t\right)=Q_{I}t=k\frac{FS_{0}}{C_{\mathrm{ATM}}}t$$

However, similar to reported vacuum-assisted pumps [9,10,11], the flow rate can be limited by the fabrication capability. From Equation (6), the flow rate is proportional to the aspect ratio which is the height of the PDMS wall over the PDMS wall thickness. For the pump with a circle shape without wings, the flow rate is written as:(7)$$Q\left(t\right)\propto \frac{S}{w}=2\pi r_{i}\frac{H}{w}$$
where, $r_{i}\;\left(m\right)$ is the inner radius of the PDMS wall, H (m) is the height of the PDMS wall, and the diffusion area is $2\;\pi r_{i}H\;\left({\mathrm{mm}}^{2}\right)$. In order to achieve large flow rates, fabricating a high aspect ratio SU-8 structure is required. One of the possible approaches is using a multiple layer process, as reported in other papers [27,28]. Another approach is to make a fractal structure as shown in this study. To be more specific, by adopting the concept of Koch snowflake structures [29] and increasing the number of wings, the flow rate can be increased within the same footprint. However, similar to the other reported vacuum-driven pumps [9,10,11,12,13], achieving the flow rates with $\mu L\cdot s^{-1}$ order still remains as a challenge, which could be studied in the future.

Collecting samples may be required in some applications for a further sample process, such as polymerase chain reaction (PCR). If needed, the proposed pump can be removed to collect samples. Since the pump is attached to the microfluidic device, peeling off or even breaking the pumping part after the operation is viable without ruining the test results. The viscosity effect of the samples is also a topic left for future study. As reported by Zhai et al. [12], high viscosity liquid takes more time to fill the channel. In this study, the viscosity effect of the sample is rounded to the empirical factor in Equation (1), whereas, further study is required to understanding the viscosity effect on the flow rates using the proposed micropump. 

## 4. Conclusions

In conclusion, we proposed a syringe-assisted, vacuum-driven micropumping device providing a constant flow rate which has the potential to be integrated into a variety of microfluidic systems. A constant flow with a rate ranging from 0.8 $\mathrm{nl}\cdot s^{-1}$ to 2.6 $\mathrm{nl}\cdot s^{-1}$ was achieved by adjusting the PDMS wall thickness from 100 μm to 200 μm, while diffusion area was fixed at 0.31 [mm^2^]. Furthermore, a constant flow rate ranging from 1.5 $\mathrm{nl}\cdot s^{-1}$ to 7.5 $\mathrm{nl}\cdot s^{-1}$ was achieved by adjusting the diffusion area from 0.31 mm^2^ to 3.35 mm^2^, while the PDMS wall thickness was fixed at 100 μm. With regards to the proposed pump, the diffusion area can be designed over a wide range by adjusting the PDMS wall height or by designing fractal structures for enlarging the lateral surface area of the PDMS wall. The concept first allows the pump to have a wide range of flow rates, as well as increasing flow rates, which has been a challenge for most passive pumps. Moreover, the proposed pump makes it easier for researchers to use their designed microfluidic devices with the proposed pump. The microfluidic researchers just need to bring their device and attach the proposed pump on their chips for the operation. As a proof-of-concept, the microfluidic channel from the other paper was chosen to demonstrate that the proposed pump has the capability to be used for predesigned microfluidic channels. A micromixing channel was adopted from the other paper and micromixing was successfully performed using the proposed pumps. As the proposed pumps are directly bonded to the outlet of the target channel, the proposed pump can selectively drive the sample in the target channel which indicates the pump has the potential to be used for complicated integrated microfluidic networks. The proposed micropump can be used instead of bulky pumps for traditional microfluidic devices.

## Figures and Tables

**Figure 1 micromachines-10-00543-f001:**
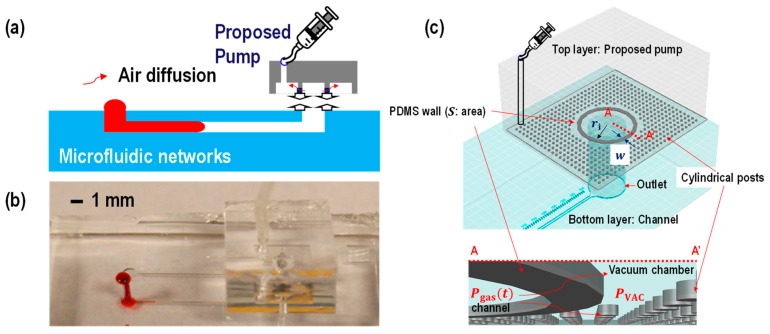
(**a**) A proposed device schematic (**b**) testing image and (**c**) three-dimensional view of the device. The bottom layer was designed to be a straight channel which is 14 mm long, 100 μm wide and 50 μm high. The top layer is designed to be a proposed pump. The inner radius of the polydimethylsiloxane (PDMS) wall ($r_{i}$) is set to be 1 mm while the PDMS wall thickness (*w*) is varied (i.e., 100 μm, 150 μm, and 200 μm) to study controlling flow rates. The height and PDMS membrane or PDMS wall shapes are also variant to study controlling flow rates. The cylindrical posts around the PDMS wall were made to avoid collapse while plungers were pulled to achieve manual vacuum condition in the vacuum chamber. Due to the gas permeability of the PDMS, the pressure difference across the PDMS membrane allows air diffusion from inside the membrane (the fluidic channel) to the outside of the PDMS membrane (the vacuum chamber).

**Figure 2 micromachines-10-00543-f002:**
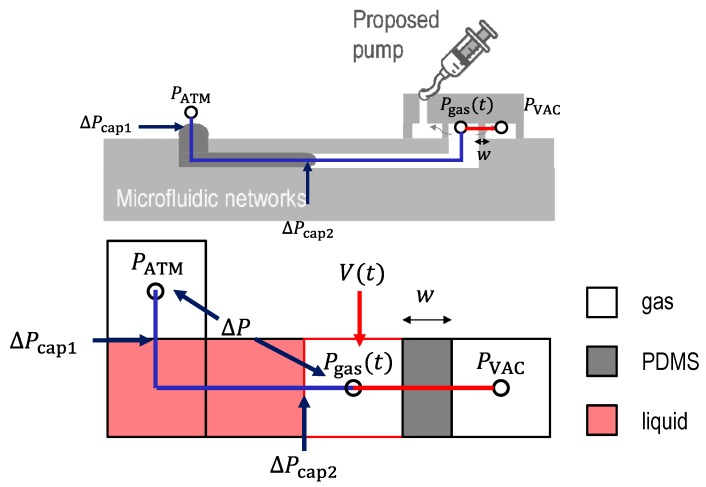
An equivalent model of the gas pressure inside channel. $P_{\mathrm{ATM}}$ (Pa) is the atmospheric pressure, $\Delta P_{\mathrm{cap}1}\;\left(\mathrm{Pa}\right)$ is the capillary pressure between gas-liquid at inlet, ΔP (Pa) is the pressure difference across the liquid column, $\Delta P_{\mathrm{cap}2}\;\left(\mathrm{Pa}\right)$ is the capillary pressure between the gas-liquid in channel, $\Delta P_{\mathrm{gas}}\left(t\right)\;\left(\mathrm{Pa}\right)$ is the gas pressure inside channel, $P_{\mathrm{VAC}}\;\left(\mathrm{Pa}\right)$ is the gas pressure inside the vacuum chamber, $V\left(t\right)\;(m^{3})$ is the gas volume inside channel, $\Delta N_{A}$ is the diffusion rate of gas molecule, w (m) is the thickness of PDMS wall, and p (Barrer) is the permeability coefficient of PDMS. Here, 1 Barrer = $3.35\times 10^{-16}\;\mathrm{mol}\cdot m^{-1\cdot }s^{-1}\cdot {\mathrm{Pa}}^{-1}$.

**Figure 3 micromachines-10-00543-f003:**
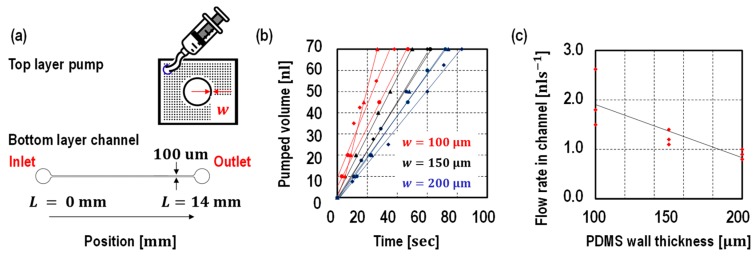
(**a**) Device schematic and (**b**) its measured pumped volume over the time, (**c**) the flow rates against the wall thickness. The PDMS wall thickness were designed to be 100 μm, 150 μm, and 200 μm. The flow time is measured corresponding to the liquid position. As shown in the graph, the pump provided a constant flow rate despite liquid position. The flow rate is inverted proportional to the PDMS wall thickness. All of the devices were tested at least three times.

**Figure 4 micromachines-10-00543-f004:**
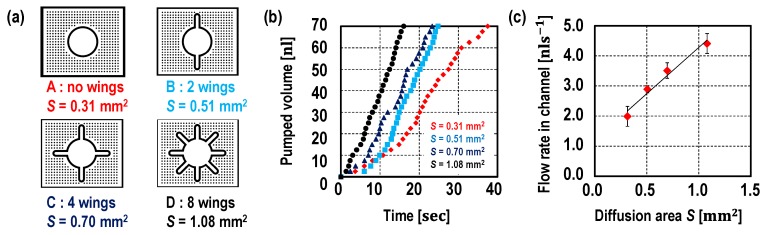
(**a**) Pump design, (**b**) measured pumped volume over the time, and (**c**) the flow rates against the diffusion areas. The diffusion areas were designed to be 0.31 mm^2^, 0.51 mm^2^, 0.70 mm^2^ and, 1.08 mm^2^, where, the PDMS wall heights were fixed at 50 μm and the PDMS wall thickness was set at 100 μm. The pumped volume while liquid flow in microfluidic channel is measured corresponding to the liquid position is shown in graph (**b**). As shown in the graph (**c**), the flow rate is proportional to the diffusion area. All of the devices were tested at least three times. Please refer to the video clips.

**Figure 5 micromachines-10-00543-f005:**
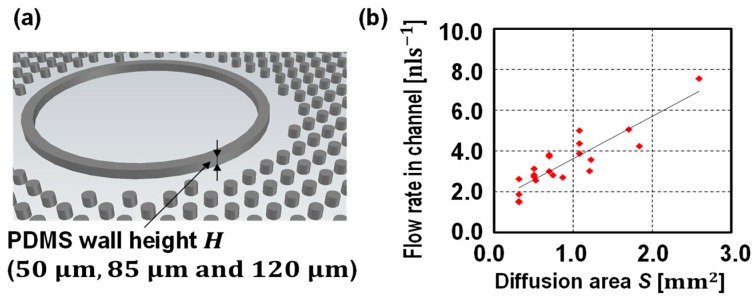
(**a**) One of the controlling diffusion area approaches and (**b**) flow rates against the diffusion area plot. The PDMS wall thickness was set at 100 μm, however, by adjusting the height of the PDMS wall (50 μm, 85 μm, and 120 μm), the diffusion areas were adjusted. Under these design conditions, the flow rates were proportional to the diffusion areas as described in Equation (1).

**Figure 6 micromachines-10-00543-f006:**
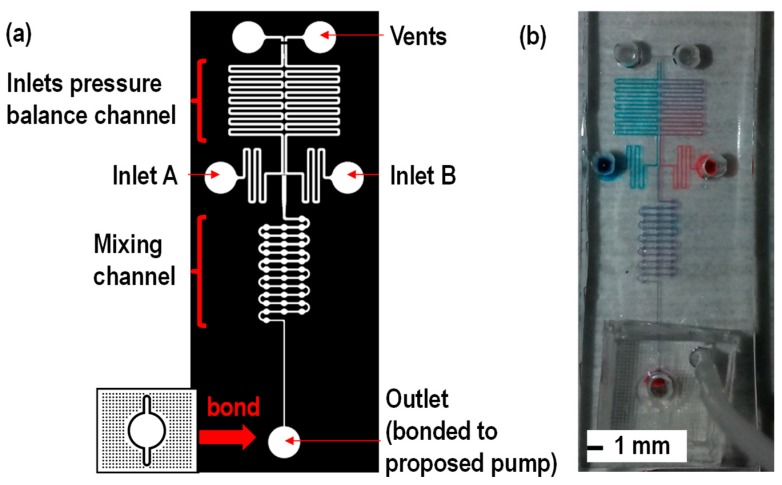
(**a**) Micromixing device channel design adopted from [12] and (**b**) experimental results. As one of the advantages of using the proposed device is that the proposed pump can be used for the operation of the device designed by [12]. In the test channel design [12], the micromixing channel design consists of two inlets, an inlet pressure balance channel for offsetting the pressure difference at inlets, the micromixing channel and the dead-end channel connected to the proposed pumping. The vent and the inlet pressure balancing channel were serving to offset the pressure difference between inlets A and B to avoid any backflow while sample mixing [12]. As shown in (**b**), micromixing of two samples was achieved with the proposed pump.

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
