# Peer review of "A Compact, Syringe-Assisted, Vacuum-Driven Micropumping Device"

_micromachines, 2019, doi:10.3390/mi10080543_

Round 1

Reviewer 1 Report

This manuscript presents efficient micro-pumping tool using air diffusion through PDMS thin layer. The pumping mechanism was very clear and applicable. It would be very promising in POCT fields. I think it is enough to be published in this journal after revisions for some comments listed below. 

1) Term "Ad-hoc" is necessary? It may make readers confused.  

2) As mentioned in equation (2) in 2.1. theory, the molar flux is proportional to (P_gas(t) - P_vac). In here, P_gas (t) is constant? (I think that P_vac could be almost constant.) flow rate could be varied as time if P_gas (t) is not constant. Of course, author showed constant flow rate (slope in figure 3 (b)). However, I think limited conditions such as the ratio of micro-channel volume/vacuum chamber volume may be required to obtain time-independent  pressure difference (P_atm - P_gas(t)). 

Author Response

Thank you for your letter and for the reviewers’ comments concerning our manuscript entitled ‘Ad-hoc syringe-assisted vacuum-driven micropumping’ (Manuscript # 578340). Those comments are all valuable and very helpful for revising and improving our paper, as well as the important guiding significance to our researches. We have studied comments carefully and have made correction which we hope meet with approval. Revised portion are marked in red in the paper. The main corrections in the paper and the responds to the reviewer’s comments are attached as a word file. 

Reviewer 2 Report

This study introduced an alternative pumping method to conventional bulky pumps in Microfluidic experiments. The paper presented the schematic of the device, explored the consistency of the pumping speed, and investigated the relationship between the shapes/areas of the outlet and the pumping speed. The experiments were well designed, the results were clearly presented.

Answer the following questions in the paper may improve this article further:

The pumped volume is at the nano liter level. Could you increase the pumping volume to microliter or milliliter level using this proposed technique? Some make-up experiments using liquid with different viscosity may be added to this study. How long does the syringe vacuum can last? How to collect the sample at the outlet? More details or discussions for the sample loading and collecting are needed.

Author Response

Thank you for your letter and for the reviewers’ comments concerning our manuscript entitled ‘Ad-hoc syringe-assisted vacuum-driven micropumping’ (Manuscript # 578340). Those comments are all valuable and very helpful for revising and improving our paper, as well as the important guiding significance to our researches. We have studied comments carefully and have made correction which we hope meet with approval. Revised portion are marked in red in the paper. The main corrections in the paper and the responds to the reviewer’s comments are as a word file.
